# Work Schedule Irregularity and the Risk of Work-Related Injury among Korean Manual Workers

**DOI:** 10.3390/ijerph17207617

**Published:** 2020-10-19

**Authors:** Won-Tae Lee, Sung-Shil Lim, Jihyun Kim, Sehyun Yun, Jin-Ha Yoon, Jong-Uk Won

**Affiliations:** 1Department of Occupational and Environmental Medicine, Severance Hospital, Yonsei University College of Medicine, Seoul 03722, Korea; lewot20@yuhs.ac (W.-T.L.); lssmail@yuhs.ac (S.-S.L.); jihyun0924@yuhs.ac (J.K.); yunsehyun@yuhs.ac (S.Y.); 2The Institute for Occupational Health, Yonsei University College of Medicine, Seoul 03722, Korea; flyinyou@yuhs.ac; 3Department of Preventive Medicine, Yonsei University College of Medicine, Seoul 03722, Korea

**Keywords:** work schedule, shift work, schedule irregularity, work-related injury

## Abstract

Work schedules comprise various variables and generate health and safety outcomes, including work-related injury, which causes socioeconomic problems, such as productivity loss and damage to worker health. We investigated the association between work schedule irregularity and the incidence of work-related injury among South Korean manual workers using data from the 5th Korean Working Conditions Survey. In total, 18,330 manual workers were included. A multivariate logistic regression analysis was performed to understand the association between work schedule and work-related injury and the influence of sufficient safety information and work schedule on work-related injury. We calculated the influence of an irregular work schedule on occupational injury after controlling for personal and work environment-related factors. The adjusted odds ratio (OR) for work-related injury was 1.66 (95% confidence interval (CI) 1.32–2.09) for an irregular work schedule. The interaction had an additive effect when the work schedule was irregular, even when sufficient safety information was provided. Manual workers had a higher incidence of injury (2.1%). Even in adjusted analyses, work schedule irregularity conferred greater risks of work injury, particularly when not working the same number of days weekly (OR 1.52, 95% CI 1.21–1.90). Policymakers and health professionals need to consider the impact of work schedule irregularity on worker safety and health.

## 1. Introduction

A work schedule comprises various variables—such as work hours, shift work, and break times between work—that are related to different health conditions, including fatigue and chronic disease. To date, numerous studies have reported the effects of shift work and working hours on workers’ health [1,2,3,4,5,6,7]. Kecklund and Axelsson reviewed the side effects of shift work with regard to hypertension, type 2 diabetes, stroke, cardiovascular disease, and breast cancer [8]. Studies have investigated the effects of work schedules on other health and safety outcomes, such as lack of sufficient sleep and stress [9,10].

Work-related injury is a global problem that affects communities and families. To prevent injuries, efforts are needed not only at the individual level but also at the organizational and governmental levels that can effect changes in policies [11]. Furthermore, injury is extremely important in occupational health; it results in an increased number of workdays lost due to injury, leading to the loss of productivity, an increase in compensation costs and medical costs, and ultimately damaging the health of workers [12].

Various factors influence work-related injuries, including age, education, gender, occupational factors, and work environments [7,13,14]. Occupational factors are estimated to cause 8.8 percent of deaths globally [15]. Manual workers are more frequently exposed to injury than other occupational groups due to ergonomic risk factors and factors associated with poor work environments [16,17,18,19]. Mining, construction, and manufacturing were the top three industries for industrial accidents [20]. According to the industrial disaster statistics of the Korea Safety and Health Agency, 94,000 cases of work-related injuries occurred in 2019, with 850 people succumbing to the injuries. As injuries are preventable, it is important to identify vulnerable groups, adopt preventive measures, and appropriately manage risks.

In this study, we aimed to analyze the association between the irregularity of work schedules and work-related injuries in Korean manual workers. Our study provides insight into these factors, which could form the foundation for the management of workplace environments and policy improvement.

## 2. Materials and Methods

### 2.1. Study Design

This study performed a secondary analysis of data from the 5th Korean Working Conditions Survey (KWCS) conducted in 2017. The survey questionnaire was developed on the basis of the European Working Conditions Survey (EWCS), conducted in Europe, and the UK Labour Force Survey [21]. The KWCS was aimed at improving worker safety and health through understanding the overall work environment of Korean workers and ascertaining their exposure to risk factors. To identify a representative sample of the economically active population, only workers aged ≥ 15 years at the time of the interview, working for wages or profits for an hour or more per week, were targeted. All the KWCS participants provided informed consent for voluntary participation; the KWCS is a nationwide open-source database that protects the anonymity and privacy of the participants. The raw data can be downloaded from the Korea Occupational Safety and Health Agency website. A previous study investigated the validity and reliability of this data [22]. In this study, we analyzed the data of only the manual worker group among the following occupational classifications: skilled workers in agriculture, forestry, and fisheries; skilled workers and related workers; equipment and machine operators; and assembly workers, simple laborers, and soldiers.

### 2.2. Outcomes and Variables

The respondents were asked if they had any health problems related to injuries sustained in the last 12 months and whether the injury was related to their work. We included only work-related injuries and excluded other injuries. During the survey, the respondents were asked about their work schedule. We evaluated whether the length of work hours was the same every week, whether the number of work days was the same every week, whether the weekly work shift was fixed, and whether the start and end time of each shift was fixed. From this information, we defined the “work schedule regularity” as a variable when all four variables were satisfied. The individual participant characteristics included gender, age, and education level, and the occupational factors included work hours, the sufficient provision of safety information, night work, and the size of the workplace. Age was classified as <39, 40–49, 50–59, or ≥60 years. The level of education was classified as elementary school or below, middle and high school, or university or higher. Work hours were divided, in accordance with the Korean Labor Standards Act, into <40, 41–52, or >53 h; the legal limit of work hours is 52 h. The size of the workplace was categorized as small (1–49 employees), medium (50–249 employees), or large (>250 employees). Out of 50,205 respondents to the KWCS, we selected 19,080 manual worker subjects. We excluded 17 individuals whose personal data were not complete. Additionally, 733 individuals who did not have work environment data related to their working hours, work regularity, and the provision of safety information were excluded. Thus, we included a study population of 18,330 manual workers with 11,266 (61.5%) men and 7064 (38.5%) women and a total of 377 work-related injuries.

### 2.3. Statistical Analysis

A frequency analysis of gender, age, final educational level, and workplace scale was conducted to show the demographic and occupational characteristics of manual workers. The differences in work-related injuries according to these characteristics were verified using the chi-square test. To understand the correlation between irregular work schedule and the dependent variable, work-related injuries, a multivariate logistic regression analysis was performed after controlling for the personal characteristics of workers and work environment-related factors. Model A involved crude analysis, Model B controlled for individual characteristics, and Model C showed the odds ratio (OR) by controlling the factors in Model B along with the workplace environment factors, such as work hours, night work, the provision of safety information, and the size of the workplace. The stratification by working hours was analyzed to determine whether working hours had an impact on the effects caused by irregular work schedules. Moreover, we analyzed the interaction between the provision of safety information and the work schedule on work-related injury. All the *p*-values are two-tailed, and *p*-values < 0.05 were considered statistically significant. We analyzed all the data using SAS version 9.4 (SAS Institute, Cary, NC, USA).

## 3. Results

Table 1 displays the relationship between the personal factors, work environment-related factors, and work-related injuries. The results revealed a statistically significant correlation between demographic characteristics, such as gender and educational level, and work-related injuries. The male manual workers experienced more injuries than their female counterparts (2.3% males vs. 1.7% female with *p* = 0.011). The prevalence of injury was 1.7%, 2.1%, and 2.8% in workers with an elementary school, middle and high school, and college or higher level of education, respectively, with *p* = 0.046. Among the work environment-related factors, work hours, the size of the workplace, the sufficient provision of safety information, night work, and regular work schedules correlated with work-related injuries. The proportion of injuries was higher among those who worked more than the legally permissible 52 h (2.8%) than among those who worked for less than 40 h (1.7%); similarly, it was higher among those who did not receive sufficient safety information (2.5%) than among those who were well informed (1.8%). The chance of injury was higher when the work schedule was irregular (2.6%) than when it was regular (1.5%), with *p* < 0.001.

Table 2 displays the influence of irregular work schedules on occupational injury after controlling for personal characteristics and work environment-related factors. In Model A, the crude ORs of work-related injury were significant, except for that in the fixed weekly work shift; we observed a similar tendency in Model B, which was controlled only for personal characteristics. Model C was controlled for personal characteristics and work environment-related factors. It revealed an adjusted OR of 1.44 (95% CI 1.16–1.80) when the daily working hours were not fixed, 1.52 (95% CI 1.21–1.90) when the weekly work shift was not fixed, and 1.37 (95% CI 1.09–1.73) when the start and end time of each shift was not fixed. The overall work schedule regularity showed an OR of 1.66 (95% CI 1.32–2.09). Thus, in the case of manual labor, the risk of work-related injuries was higher in those with irregular work schedules.

Table 3 displays the results of the stratification analysis for working hours. All the analyses were adjusted for all personal characteristics and work environment-related factors, except working hours. The risk of irregularity was higher in subgroups working for more than 52 h (OR 2.61, 95% CI 1.33–2.38) than in those working for less than 52 h (OR 2.05, 95% CI 1.52–2.76). The results for the remaining factors were similar to those in the non-stratification analysis.

We analyzed the interaction to see how it affected work-related injury depending on the provision of safety information and the regularity of work schedules (Figure 1). Compared with a reference group with a regular schedule and sufficient safety information provided, the risk was statistically significant when the work schedule regularity was irregular and sufficient safety information was present, and it was higher when safety information was not provided. Fixed daily working hours (OR 1.78, 95% CI 1.33–2.38), fixed weekly working days (OR 1.85, 95% CI 1.37–2.51), and having a fixed start and end time for each shift (OR 1.69, 95% CI 1.27–2.24) showed a similar tendency for work-related injuries as that of work schedule regularity (OR 2.02, 95% CI 1.50–2.72); however, in the case of the fixity of the weekly work shift, the OR was not significant for differences between the provision and non-provision of sufficient safety information.

## 4. Discussion

In this study, we investigated the effects of the irregularity of work schedules on the risk of work-related injuries among manual workers. In 2018, 0.48% of all Korean workers suffered occupational injuries. Manual workers, such as those in construction and manufacturing, are more likely to be injured than other workers [23,24,25]. In our study, the incidence of work-related injuries was 2.1% over 12 months in the manual workers group, indicating that they have a higher likelihood of sustaining injuries. With regard to personal characteristics, occupational injuries were common in men and less educated groups (Table 1), as reported previously [23,25]. In Australia, most of the injuries in blue-collar workers and those in small workplaces were visible and unmanaged [18]. In the case of work environment-related factors, injuries were higher when the work hours exceeded 52 h (Table 3) [26]. This is consistent with the results of a study that found that longer work hours increased the rate of industrial accidents [27,28]. A prospective study on construction workers in the USA determined that the risk of severe work-related injury increased (OR 1.98) when work hours exceeded 50 h per week [29]. Many studies have shown that work-related injuries occur more frequently in shift work than in non-shift work. When a worker changed from day work to non-standard shift work, the OR for occupational injuries increased to 2.6 (95% CI 1.79–3.77) [2,27]. The results of the logistic regression analysis (Table 2) suggested that irregular work schedules induce a greater risk of work injuries, even when personal and other occupational factors, such as work hours, are controlled. In particular, the number of workdays in a week correlated with the highest risk of occupational injuries (OR 1.52, 95% CI 1.21–1.90). We conducted the stratification analysis assuming working hours as a confounder, but there was no change in the results (Table 3). Thus, the weekly working hours did not influence the association between the irregularity of the work schedule and the incidence of injury. According to Kang et al., working hours can modify the risk of workplace injury [28], but the cross-sectional nature of our research design could not measure these socio-biological selection biases, so-called health worker effects [30].

Moreover, the irregularity of work schedules and the provision of safety information had an additive effect on occupational injury. Our findings indicate that the risk of injury was greater when sufficient safety information was not provided (Figure 1) [31,32,33]. Furthermore, among the variables, the highest risk was noted for the same number of workdays per week with the variation in total schedule regularity. This suggests that if the number of workdays per week varies, the probability of working on a weekend increases, and, consequently, time lost from work due to occupational injury increases [34,35]. In particular, in women the percentage of lost time due to injury on Sundays or Saturdays increased by approximately 122% and 60%, respectively [34]. This may be caused by differences in the level of supervision on the weekends as several other events occur, such as social activities and late dining; additionally, alcohol consumption may increase on the weekends [36]. Moreover, working different hours each week indicate a fluctuation in job demand, which can directly cause work-related injury; as the job demand increases, injuries can occur through mechanisms that affect job-related stress and musculoskeletal disorders [17,37].

The strength of our research is that it analyzes, for the first time, the relationship between work injuries on a large scale and irregularities in work schedules in manual workers. The study utilized well-established surveys for the entire population and analyzed the occupational factors for health after controlling for individual characteristics. In addition, our study has the advantage that the risk factors were analyzed using the regularity of the work schedule, which is applicable to workers who were not performing shift work.

This study has its limitations. The data used in this study are from a cross-sectional survey, and therefore the prognostic relationship cannot be identified in the relationship between the irregularity of the work schedule and work-related injury. Therefore, further prospective research is needed to clarify causality in this context. However, the data are relatively robust, as they were sampled from all categories of workers, including manual laborers, throughout the country. Several human and environmental factors influence the risk of injury: unsafe acts, health conditions, physical (ergonomic) and chemical hazards of the workplace, and job-related stress [38,39]. We adjusted other confounders available in this dataset, but because of the data limitations, various other factors could not be analyzed. Nonetheless, in this study we confirmed that an irregular work schedule influences work-related injuries even after adjusting for working hours and shift work, which are important factors that cause injury. In addition, when measuring injury, the data were inadequate to ascertain the exact mechanism of injury because the data were based on self-reports, which are not objective. Despite that, the severe occupational injuries that we wanted to investigate would not be overestimated without bias. Notwithstanding these limitations, our results directly address the suspected risk factors that may be related to workplace injuries.

## 5. Conclusions

We analyzed the influence of the regularity of work schedules on work-related injuries after controlling for personal and work environment-related factors among manual workers. Even when the work hours and work environments, which have recently been a topic of discussion, were consistent, our results confirmed that an irregular work schedule is the main factor that affects worker safety and health. Thus, these analyses provide useful insight for improving the work environment for the health promotion of workers. Policymakers need to consider how irregular work schedules affect worker safety and health.

## Figures and Tables

**Figure 1 ijerph-17-07617-f001:**
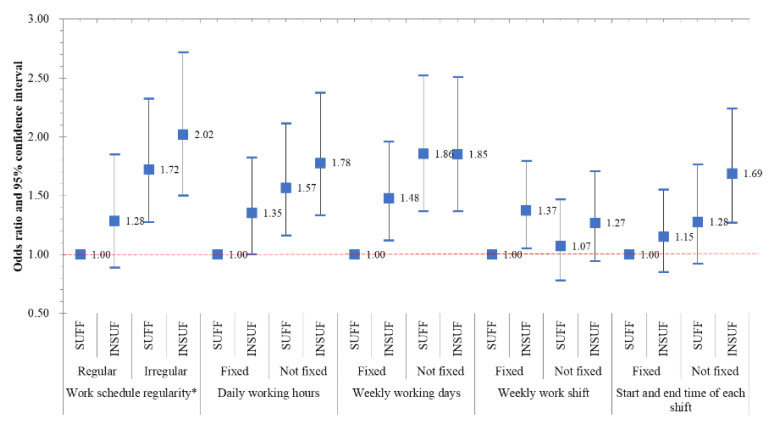
Odds ratio of work-related injuries according to work schedule regularity with sufficiently and insufficiently provided safety information. Work schedule regularity * is regular when the daily working hours, weekly working days, weekly work shift, and start and end time of each shift are all “fixed”. “SUFF” means that sufficient safety information was provided. All the models were controlled for age, gender, education level, the size of the workplace, and work hours.

**Table 1 ijerph-17-07617-t001:** General characteristics of the study population and the status of work-related injury according to individual and occupational characteristics.

Characteristics	Total		Work-Related Injury				*p*-Value
			Yes		No		
	*n*	%	*n*	%	*n*	%	
Gender							0.011
Male	11,266	(61.5)	256	(2.3)	11,010	(97.7)	
Female	7064	(38.5)	121	(1.7)	6943	(98.3)	
Age (years)							0.369
≤39	2349	(12.8)	40	(1.7)	2309	(98.3)	
40–49	2887	(15.8)	66	(2.3)	2821	(97.7)	
50–59	4901	(26.7)	109	(2.2)	4792	(97.8)	
≥60	8193	(44.7)	162	(2.0)	8031	(98.0)	
Educational level							0.046
Elementary school	4176	(22.8)	104	(2.5)	4072	(97.5)	
Middle and high school	11,027	(60.2)	220	(2.0)	10,807	(98.0)	
College	3127	(17.1)	53	(1.7)	3074	(98.3)	
Work hours (h)							<0.001
≤40	9166	(50.0)	157	(1.7)	9009	(98.3)	
41–52	4902	(26.7)	102	(2.1)	4800	(97.9)	
≥53	4262	(23.3)	118	(2.8)	4144	(97.2)	
Size of the workplace							0.006
Small	15,648	(85.4)	343	(2.2)	15,305	(97.8)	
Middle	1484	(8.1)	16	(1.1)	1468	(98.9)	
Large	1198	(6.5)	18	(1.5)	1180	(98.5)	
Provision of safety information							0.003
Sufficient	10,588	(57.8)	189	(1.8)	10,399	(98.2)	
Insufficient	7742	(42.2)	188	(2.4)	7554	(97.6)	
Night work							0.436
Day work	16,482	(89.9)	344	(2.1)	16,138	(97.9)	
Night work	1848	(10.1)	33	(1.8)	1815	(98.2)	
Work schedule regularity *							<0.001
Regular	8667	(47.3)	128	(1.5)	8539	(98.5)	
Irregular	9663	(52.7)	249	(2.6)	9414	(97.4)	
Daily working hours							<0.001
Fixed	10,927	(59.6)	186	(1.7)	10,741	(98.3)	
Not fixed	7403	(40.4)	191	(2.6)	7212	(97.4)	
Weekly working days							<0.001
Fixed	12,093	(66.0)	210	(1.7)	11,883	(98.3)	
Not fixed	6237	(34.0)	167	(2.7)	6070	(97.3)	
Weekly work shift							0.518
Fixed	11,451	(62.5)	229	(2.0)	11,222	(98.0)	
Not fixed	6879	(37.5)	148	(2.2)	6731	(97.8)	
Start and end time of each shift							<0.001
Fixed	11,417	(62.3)	198	(1.7)	11,219	(98.3)	
Not fixed	6913	(37.7)	179	(2.6)	6734	(97.4)	
Total	18,330	(100.0)	377	(2.1)	17,953	(97.9)	

Work schedule regularity * is “regular” when the daily working hours, weekly working days, weekly work shift, and start and end time of each shift are all “fixed”.

**Table 2 ijerph-17-07617-t002:** Odds ratio and 95% confidence intervals for work-related injury in models A, B, and C.

Variables	Model A		Model B		Model C	
	ORs	95% CIs	ORs	95% CIs	ORs	95% CIs
Work schedule regularity *						
Regular	1.00		1.00		1.00	
Irregular	1.77	1.42–2.19	1.73	1.39–2.17	1.66	1.32–2.09
Daily working hours						
Fixed	1.00		1.00		1.00	
Not fixed	1.53	1.25–1.88	1.52	1.22–1.88	1.44	1.16–1.80
Weekly working days						
Fixed	1.00		1.00		1.00	
Not fixed	1.56	1.27–1.91	1.53	1.23–1.91	1.52	1.21–1.90
Weekly work shift						
Fixed	1.00		1.00		1.00	
Not fixed	1.08	0.87–1.33	1.02	0.82–1.27	0.99	0.79–1.24
Start and end time of each shift						
Fixed	1.00		1.00		1.00	
Not fixed	1.51	1.23–1.85	1.53	1.22–1.91	1.37	1.09–1.73

Work schedule regularity * is regular when the daily working hours, weekly working days, weekly work shift, and start and end time of each shift are all “fixed”. Model A: unadjusted; Model B: adjusted for age, gender, and education; and Model C: adjusted for age, gender, education, work hours, night work, the provision of safety information, and the size of the workplace. OR, odds ratio; CI, confidence interval.

**Table 3 ijerph-17-07617-t003:** Odds ratio and 95% confidence intervals for work-related injury in stratification analysis for working hours.

Variables	Working Hours (≤52 h)		Working Hours (>52 h)	
	ORs	95% CIs	ORs	95% CIs
Work schedule regularity *				
Regular	1.00		1.00	
Irregular	2.05	1.52–2.76	2.61	1.64–4.16
Daily working hours				
Fixed	1.00		1.00	
Not fixed	1.62	1.23–2.12	1.59	1.05–2.42
Weekly working days				
Fixed	1.00		1.00	
Not fixed	1.81	1.38–2.38	2.06	1.34–3.15
Weekly work shift				
Fixed	1.00		1.00	
Not fixed	1.06	0.81–1.39	0.99	0.79–1.24
Start and end time of each shift				
Fixed	1.00		1.00	
Not fixed	1.47	1.01–1.96	1.54	0.97–2.44

Work schedule regularity * is regular when the daily working hours, weekly working days, weekly work shift, and start and end time of each shift are all “fixed”. Adjusted for age, gender, education, night work, the provision of safety information, and the size of the workplace. OR, odds ratio; CI, confidence interval.

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
