# Peer review of "Work Schedule Irregularity and the Risk of Work-Related Injury among Korean Manual Workers"

_ijerph, 2020, doi:10.3390/ijerph17207617_

Round 1
Reviewer 1 Report
First of all, I would like to congratulate the authors for the manuscript and the current work. I consider that the manuscript is interesting, however, there are some areas that need work:
- The introduction needs improvement. This section does not provide enough information regarding the topic and the country studied.
- The materials and methods does to include how the authors had access to the data. Please provide this information
- The results are highly interesting but the tables and figure are misplaced (Table 2) or too complicated to understand(Table 1). Also, I recommend further explanation of the results for the data since the readers may get lost with so much information.

Author Response
Responses to the comments of Reviewer #1
Reviewer Point P 1 – The introduction needs improvement. This section does not provide enough information regarding the topic and the country studied.
(Response) We agreed with the reviewer on this important point. In the introduction, we focused more on the injuries of workers and the country we studied. We added the following explanation.
Line 48 “Occupational factors are estimated to cause 8.8 percent of deaths globally [15]. Manual workers are more frequently exposed to injury than other occupational groups are due to ergonomic risk factors and factors associated with poor work environments [16–19]. Mining, construction, and manufacturing were the top three industries for industrial accidents [20]. According to the industrial disaster statistics of the Korea Safety and Health Agency, 94,000 cases of work-related injuries occurred in 2019, with 850 people succumbing to the injuries.”
Reviewer Point P 2 – The materials and methods does to include how the authors had access to the data. Please provide this information
(Response) We feel sorry for not describing how we had access to the database. We added and edited our methods section as your comment, as below. Please read our revised manuscript too.
Line 69 “All KWCS participants provided informed consents for voluntary participation; the KWCS is a nationwide open-source database that protects the anonymity and privacy of the participants. The raw data can be downloaded from the Korea Occupational Safety and Health Agency website.”
Reviewer Point P 3 – The results are highly interesting but the tables and figure are misplaced (Table 2) or too complicated to understand (Table 1). Also, I recommend further explanation of the results for the data since the readers may get lost with so much information.
(Response) – Thank you for your constructive comment. The authors fully agreed with the reviewer’s opinion. We replace Tables 1, 2, and Figure 1 to following logical flow. The following logic is the first demographic characteristics according to injury experience (Table 1), and next is crude and multivariate logistic regression (Table 2), and the interactive association is placed in the end (figure 1). We also remove the odds ratio with 95% CI in Table 1, which is duplicated in Table 2 of Model A. and we added and revised the results to make it easier to understand.
Line 110 “The male manual workers experienced more injuries than their female counterparts (2.3 % males vs. 1.7% female with p = 0.011). The prevalence of injury is 1.7%, 2.1%, and 2.8% in workers with elementary school, middle and high school, and college or higher level of education, respectively, with p = 0.046. Among the work environment-related factors, work hours, size of the workplace, sufficient provision of safety information, night work, and regular work schedules correlated with work-related injuries. The proportion of injuries was higher among those who worked more than the legally permissible 52 h (2.8%) than among those who worked for less than 40 h (1.7%); similarly, it was higher among those who did not receive sufficient safety information (2.5%) than those who were well informed (1.8%). The chance of injury was higher when the work schedule was irregular (2.6%) than when it was regular (1.5%) with p < 0.001.”

Reviewer 2 Report
It is an interesting study to look at the association between work irregularity and occupational injury.
1. The major concern of this study is that the working hours of workers with irregular schedule were not equal to those with regular one because the definition of study subjects is workers who works one or more hours per week. It is natural that occupational injuries are common in workers who are working longer times. There are many studies about numbers of occupational injuries based on working time such as Kang, S.-K. Kwon O.-J., Occupational Injury Statistics in Korea, Saf Health Work, 2(1), 2011: 52-56. Thus, authors must show the similarity of average or total working hours between the group of Regular and Irreglar schedule. If the irregular scheduled workers work longer hours than regular workers, it is a natural result that occupational injuries are high in the group. Also think about FTE (full time equivalent) when you compare occupational injuries among different groups.
2. The merit of using Korean Working Condition Survey is a well-designed sampling. It should be described or referred. for example as follows. 1) Seo J.H. Development of a Tailored Analysis System for Korean Working Conditions Survey, Saf Health Work, 3 (2016), 201-207; 2) Y.S. Kim, K.Y. Rhee, M.J. Oh, J. ParkThe validity and reliability of the second Korean Working Conditions Survey. Saf Health Work, 4 (2013), pp. 111-116.
Author Response
Responses to the comments of Reviewer #2
Thank you for taking the time to read our original manuscript and to provide feedback. We have enclosed a point by point list of changes or rebuttals along with our revised manuscript.
Reviewer Point P 1 – The major concern of this study is that the working hours of workers with irregular schedule were not equal to those with regular one because the definition of study subjects is workers who works one or more hours per week. It is natural that occupational injuries are common in workers who are working longer times. There are many studies about numbers of occupational injuries based on working time such as Kang, S.-K. Kwon O.-J., Occupational Injury Statistics in Korea, Saf Health Work, 2(1), 2011: 52-56. Thus, authors must show the similarity of average or total working hours between the group of Regular and Irreglar schedule. If the irregular scheduled workers work longer hours than regular workers, it is a natural result that occupational injuries are high in the group. Also think about FTE (full time equivalent) when you compare occupational injuries among different groups.
(Response) Thank you for your constructive comment. We agreed that high exposure (long working hours) generally at risk of injury. So, the long working hours may role as effect modifier beyond confounding factor. Although we adjusted working hours as confounding factor in logistic regression model, the stratification analysis is needed to clarify the association as your comment. So, we undertook stratification analysis for sensitivity analysis, as below.
There is no change in direction over the working hours of stratification analysis. Thus, there is little interaction of weekly working hours in the association between the regularity of the work schedule and the experience of injury. Nevertheless, our sensitivity analysis cannot explain biological and sociological interactions, such as the effects of health workers' survival because they are based solely on statistical interactions. These are some of the main limitations of current research. We added a sentence about it to the discussion section.
Line 140 “Table 3 displays the results of the stratification analysis for working hours. All analyses were adjusted for all personal characteristics and work environment-related factors except working hours. The risk of irregularity was higher in subgroups working for more than 52 h (OR 2.61, 95% CI 1.33–2.38) than those working for less than 52 h (OR 2.05, 95% CI 1.52–2.76). The results for the remaining factors were similar to those in the non-stratification analysis.”
Table 3. Odds ratio and 95% confidence intervals for work-related injury in stratification analysis for working hours.
|
Variables |
Legal working hours (≤52 h) |
Long working hours (>52 h) |
||
|
ORs |
95% CIs |
ORs |
95% CIs |
|
|
Regularity |
|
|
|
|
|
Regular |
1.00 |
– |
1.00 |
– |
|
Irregular |
2.05 |
1.52–2.76 |
2.61 |
1.64–4.16 |
|
Same number of hours/day |
|
|
|
|
|
Regular |
1.00 |
– |
1.00 |
– |
|
Irregular |
1.62 |
1.23–2.12 |
1.59 |
1.05–2.42 |
|
Same number of days/week |
|
|
|
|
|
Regular |
1.00 |
– |
1.00 |
– |
|
Irregular |
1.81 |
1.38–2.38 |
2.06 |
1.34–3.15 |
|
Fixed weekly work shift |
|
|
|
|
|
Fixed |
1.00 |
– |
1.00 |
– |
|
Not fixed |
1.06 |
0.81–1.39 |
0.99 |
0.79–1.24 |
|
Fixed starting and finishing times |
|
|
|
|
|
Fixed |
1.00 |
– |
1.00 |
– |
|
Not fixed |
1.47 |
1.01–1.96 |
1.54 |
0.97–2.44 |
Adjusted for age, gender and education, night work, provision of safety information, and size of the workplace. OR, odds ratio; CI, confidence interval.
Line 182 “We conducted the stratification analysis assuming working hours as a confounder, but there was no change in the results (Table 3). Thus, the weekly working hours did not influence the association between the irregularity of the work schedule and the incidence of injury. According to Kang et al., working hours can modify the risk of workplace injury [28], but the cross-sectional nature of our research design could not measure these socio-biological selection biases, so-called health worker effects [30].”
Reviewer Point P 2 – The merit of using Korean Working Condition Survey is a well-designed sampling. It should be described or referred. for example as follows. 1) Seo J.H. Development of a Tailored Analysis System for Korean Working Conditions Survey, Saf Health Work, 3 (2016), 201-207; 2) Y.S. Kim, K.Y. Rhee, M.J. Oh, J. ParkThe validity and reliability of the second Korean Working Conditions Survey. Saf Health Work, 4 (2013), pp. 111-116.
(Response) We entirely agreed with the reviewer’s opinion. We attached the merit of using this data and a reference to it.
Line 71 “A previous study investigated the validity and reliability of this data [22].”

Reviewer 3 Report
Some considerations:
- Positive aspects of the paper - the number of workers is representative and the results, in general, are in accordance with anothers researches around the world.
- But, there are a lot of anothers characteristics that can be involved for the influence the manual workers injurie´s incidence in relation of the regularity of work schedule. Therefore, It´s recommended to emphatizes that difficulties to confirm tottaly the results of the paper.
Author Response
Responses to the comments of Reviewer #3
Thank you for taking the time to read our original manuscript and to provide feedback. We have enclosed a point by point list of changes or rebuttals along with our revised manuscript.
Reviewer Point P 1 – Positive aspects of the paper - the number of workers is representative and the results, in general, are in accordance with anothers researches around the world. But, there are a lot of anothers characteristics that can be involved for the influence the manual workers injurie´s incidence in relation of the regularity of work schedule. Therefore, It´s recommended to emphatizes that difficulties to confirm totally the results of the paper.
(Response) Thank you for your detailed comments. We highlight the work schedule irregularity effect on injury risk in the current study. However, as your comment, injury epidemiology is known to be influenced by various variables in other studies. So, we added limitation section, as below.
Line 213 “Several human and environmental factors influence the risk of injury: unsafe acts, health conditions, physical(ergonomic) and chemical hazards of workplace, and job-related stress [39,40]. We adjusted other confounders available in this dataset, but because of the data limitations, various other factors could not be analyzed. Nonetheless, in this study, we confirmed that the irregular work schedule influences work-related injuries even after adjusting for working hours and shift work, which are important factors that cause injury.”

Round 2
Reviewer 1 Report
I think that the authors have addressed all my comments perfectly and I find the currently manuscript highly appealing.
Author Response
Thank you again for your thoughtful comments.
Reviewer 3 Report
The author´s review is adequate and confirmed that the irregular work schedule influences could be an important factor that cause injury.
Author Response

(The authors gave the same response as above.)
